

# Source apportionment of organic aerosol from two-year highly time-resolved measurements by an aerosol chemical speciation monitor in Beijing, China

Yele Sun[1,2,3], Weiqi Xu[1,2], Qi Zhang[4], Qi Jiang[1*], Francesco Canonaco[5], André S. H. Prévôt[5], Pingqing

Fu[1,2,3], Jie Li[1], John Jayne[6], Douglas R. Worsnop[6], and Zifa Wang[1,2,3]

[1]State Key Laboratory of Atmospheric Boundary Layer Physics and Atmospheric Chemistry, Institute of Atmospheric Physics, Chinese Academy of Sciences, Beijing 100029, China
[2]University of Chinese Academy of Sciences, Beijing 100049, China
[3]Center for Excellence in Regional Atmospheric Environment, Institute of Urban Environment, Chinese Academy of
Sciences, Xiamen 361021, China
[4]Department of Environmental Toxicology, University of California, 1 Shields Ave., Davis, CA 95616, USA
[5]Laboratory of Atmospheric Chemistry, Paul Scherrer Institute, Villigen PSI 5232, Switzerland
[6]Aerodyne Research, Inc., Billerica, MA 01821, USA
*Now at National Meteorological Centre, Beijing 100081, China

*Correspondence to*: Yele Sun (sunyele@mail.iap.ac.cn)

**Abstract.** Organic aerosol (OA) represents a large fraction of submicron aerosols in the megacity of Beijing, yet long-term characterization of its sources and variations is very limited. Here we present analysis of in situ measurements of OA in submicrometer particles with an aerosol chemical speciation monitor (ACSM) for two years from July 2011 to May 2013. The sources of OA are analyzed with multilinear engine (ME-2) by constraining three primary OA factors including fossil

fuel related OA (FFOA), cooking OA (COA), and biomass burning OA (BBOA). Two secondary OA (SOA), representing a less oxidized oxygenated OA (LO-OOA) and a more oxidized (MO-OOA) are identified during all seasons. The monthly average concentration OA varied from 13.6 to 46.7 µg m$^{-3}$ with a strong seasonal pattern that is usually highest in winter and lowest in summer. FFOA and BBOA show similarly pronounced seasonal variations with much higher concentrations and contributions in winter due to enhanced coal combustion and biomass burning emissions. The contribution of COA to OA,

however, is relatively stable (10 – 15%) across different seasons, yet presents significantly higher values at low relative humidity levels (RH < 30%), highlighting the important role of COA during clean periods. The two SOA factors present very different seasonal variations. The pronounced enhancement of LO-OOA concentrations in winter indicates that emissions form combustion-related primary emissions could be a considerable source of SOA under low temperature (*T*) conditions. Comparatively, MO-OOA shows high concentrations consistently at high RH levels across different *T* levels, and

the contribution of MO-OOA to OA is different seasonally with lower values occurring more in winter (30 – 34%) than other seasons (47 – 64%). Overall, SOA (= LO-OOA + MO-OOA) dominates OA composition during all seasons by contributing 52 – 64% of the total OA mass in heating season, and 65 – 75% in non-heating seasons. The variations of OA composition




as a function of OA mass loading further illustrates the dominant role of SOA in OA across different mass loading scenarios during all seasons. However, we also observed a large increase in FFOA associated with a corresponding decrease in MO-OOA during periods with high OA mass loadings in heating season, illustrating an enhanced role of coal combustion emissions during highly polluted episodes. Potential source contribution function analysis further shows that the transport

from the regions located to the south and southwest of Beijing within ~200 km can contribute substantially to high FFOA and BBOA concentrations in heating season.

## 1    Introduction

Organic aerosol (OA) is ubiquitous in the atmosphere and constitutes a large fraction of submicron aerosol worldwide (Zhang et al., 2007;Jimenez et al., 2009). Organic aerosol has two different sources. One is direct emissions from

combustion processes (e.g., burning of fossil fuel and biomass), which is known as primary organic aerosol (POA). The other is secondary formation from the oxidation of volatile organic compounds (VOCs) termed as secondary organic aerosol (SOA) (Hallquist et al., 2009). Recent studies have found that SOA is the major component of OA not only in rural and remote regions, but also in the highly polluted urban regions (Zhang et al., 2007;Jimenez et al., 2009). Although model simulations of SOA have been improved during the last decade attributing to significant improvements in understanding the

formation mechanisms and volatility of OA, the discrepancy between model simulations and ambient observations can still be substantial (Shrivastava et al., 2011;Fu et al., 2012;Fast et al., 2014). As a result, SOA contributes one of the largest uncertainties in evaluating climate radiation forcing of aerosol particles (Boucher et al., 2013). While a better understanding of SOA formation and evolutionary mechanisms is essential to improve model performances (Shrivastava et al., 2017), constraining the models with observations, particularly long-term measurements would be one of the most effective ways to

reduce the radiative forcing uncertainties.

Aerodyne aerosol mass spectrometer (Jayne et al., 2000;Canagaratna et al., 2007) is one of the state-of-the-art instruments by providing quantative measurement of organic aerosol in real-time. Subsequent analysis of OA mass spectra using receptor models, e.g., multiple component analysis (Zhang et al., 2005), positive matrix factorization (PMF, Ulbrich et al., 2009) and multilinear engine (Canonaco et al., 2013) can further differentiate various OA factors from different sources and processes.

Since 2006, AMS has been widely deployed at various regions in China for real-time characterization of non-refractory submicron aerosol (NR-PM$_1$) species (Li et al., 2017b and references therein). The sources of OA in different seasons were analyzed using PMF. While hydrocarbon-like OA (HOA) and oxygenated OA (OOA) are ubiquitously identified, cooking OA (COA) in urban areas, biomass burning OA (BBOA) and coal combustion OA (CCOA) in specific seasons are also resolved. PMF analysis of high resolution OA mass spectra can further differentiate different types of SOA, for instance, less

oxidized OOA (LO-OOA) and more oxygenated OOA (MO-OOA). However, most of these studies are focused on intensive measurements in relatively short periods (e.g., 1 -2 months) in a single season (mostly in either summer or winter), long-term



measurements and characterization of OA in China are still rather limited. Zhang et al. (2013) reported the season variations of NR-PM$_1$ species in Beijing in 2008. The results showed the dominance of OA during all seasons. Similar conclusions were also found from the high resolution AMS (HR-AMS) measurements from 2012 to 2013 in Beijing (Hu et al., 2017). While both studies showed the dominant role of OOA in summer, the contribution of POA was quite different (76% vs.

50%). However, most measurements in the these two studies lasted approximately one month in each season, and our understanding of the sources and variations of OA is far from complete, particularly in a city with frequent changes in different air masses and largely different emissions sources (Guo et al., 2014;Sun et al., 2015;Zheng et al., 2016). Sun et al. (2015) conducted one year real-time measurements of NR-PM$_1$ species from 2011 – 2012 using an aerosol chemical speciation monitor (ACSM). While OA showed a similar season variation as those in Zhang et al. (2013) and Hu et al.

(2017), substantial differences in monthly averaged mass concentrations were also observed, highlighting the importance of continuous real-time measurements  for understanding seasonal characteristics. However, since this study only presents the characterization of the total OA, the sources and seasonal variations of different OA factors remain less understood.

In this work, we present an analysis of nearly two years' measurements of OA by an ACSM in the megacity of Beijing. Although year - round measurements of NR-PM$_1$ species and source characterization of OA have been reported at many sites

throughout the globe (Minguillón et al., 2015;Parworth et al., 2015;Petit et al., 2015;Ripoll et al., 2015;Bressi et al., 2016;Schlag et al., 2016;Rattanavaraha et al., 2017),  real-time source characterization of OA in Beijing for more than one year is never reported. Here the sources of OA in each season are determined by the bilinear model with multilinear engine (ME-2) (Paatero, 1999). The seasonal variations, diurnal cycles, and relative humidity and temperature dependence of OA source factors are elucidated. The roles of different OA factors, and POA and SOA in haze pollution are discussed, and the

potential source regions are also investigated with potential source contribution function analysis. To our knowledge, this study presents the longest continuous characterization of OA in Beijing, which is of great importance for validating and constraining the chemical transport models.

## 2    Experimental methods

### 2.1  Sampling

An Aerodyne Aerosol Chemical Speciation Monitor (Ng et al., 2011b) was deployed at an urban site, the tower branch of Institute of Atmospheric Physics (39°58′N, 116°22′E; ASL: 49 m), in Beijing for a long-term real-time measurements of NR-PM$_1$ species including organics (Org), sulfate (SO$_4$), nitrate (NO$_3$), ammonium (NH$_4$) and chloride (Chl). The measurements were conducted for nearly two years from July 2011 to May 2013 with a time resolution of approximately 15 min. In addition, gaseous species of NO$_y$, NO, and O$_3$ were simultaneously measured with a suite of gas analyzers during the

same period, while CO and SO$_2$ were measured during the period of December 2011 – May 2013. Black carbon (BC), light



extinction of dry particles at 630 nm, gaseous $NO_2$ were also measured from August 2012 to May 2013 using an Aethalometer (AE22, Magee Scientific), a Cavity Attenuated Phase Shift (CAPS) extinction monitor, and a CAPS $NO_2$ monitor (Kebabian et al., 2008), respectively. The meteorological parameters, temperature (*T*) and relative humidity (RH) at ground site, and wind direction (WD) and wind speed (WS) at 240 m were obtained from the measurements on the Beijing
325 m meteorological tower. More detailed descriptions of the sampling site, the operations of the ACSM, and the collocated measurements are given elsewhere (Sun et al., 2012;Ge et al., 2013;Sun et al., 2015;Wang et al., 2015).

## 2.2  Data Analysis

The ACSM data were analyzed using the standard software (v.1.5.3.0) written in Igor Pro (WaveMetrics, Inc., Oregon USA). The mass concentrations of NR-PM$_1$ species, and the mass spectra of OA (*m/z* 12 – 140) were determined using a
composition-dependent collection efficiency recommended by Middlebrook et al. (2012) and default relative ionization efficiencies (RIE) except ammonium that was determined from pure ammonium nitrate. More detailed evaluations on the mass quantifications are given elsewhere (Sun et al., 2012;Sun et al., 2013b).

Positive matrix factorization was first performed to the ACSM OA mass spectra (*m/z* 12 – 120) in each season. The detailed procedures for pretreatment of data and error matrices have been given in Ng et al. (2011b) and Sun et al. (2012). As shown
in Fig. S1, two factors, i.e., a hydrocarbon-like OA (HOA) and an oxygenated OA (OOA) can be relatively well resolved during all seasons. Although extending the PMF solution to more factors can help to reduce uncertainties in separation of POA and SOA, e.g., four factors in winter 2011-2012 (Sun et al., 2013b), it often generates unrealistic factors due to the low sensitivity, mass resolution, and limited *m/z*'s of the ACSM measurements. Moreover, it is very challenging for PMF-ACSM to separate BBOA from other POA factors, traffic-related HOA from COA in summer, and different types of SOA in winter
according to our previous studies (Sun et al., 2012;Sun et al., 2013b;Zhang et al., 2016). Therefore, the multilinear engine algorithm (ME-2) (Paatero, 1999) by using the prior known source information as constraints was used for source apportionment of OA.

In this work, the a-value approach was used for ME-2 analysis (Canonaco et al., 2013). The mass spectral profiles of three primary OA factors, i.e., standard HOA and BBOA from Ng et al. (2011a) and COA from Sun et al. (2016b) were
constrained by varying a-values from 0 to 1, while the other factors were left free. ME-2 analysis was first performed to the entire dataset assuming that all OA factors have similar spectral profiles in different seasons. Such an assumption could introduce large uncertainties for secondary aerosol factors considering the large differences in meteorological conditions and precursors of VOCs in different seasons. We then performed ME-2 analysis on the seasonal datasets of ACSM, which include summer 2011 (S11), fall 2011 (F11), winter 2011 (W11), spring 2012 (Sp12), summer 2012 (S12), fall 2012 (F12),
winter 2012 (W12), and spring 2013 (Sp13). Indeed, the mass spectral profiles of LO-OOA from the seasonal ME2-ACSM





analysis show large differences in different seasons (Fig. 1). Therefore, the results from the seasonal ME2-ACSM analysis are used for the discussions. To better compare the variations in primary and secondary OA in different seasons, and also allow for some degrees of freedom for model runs, the five-factor solution, i.e., fossil fuel related OA (FFOA), COA, BBOA, and a less oxidized OOA (LO-OOA) and a more oxidized OOA (MO-OOA), from the average of three model runs

with a-values of 0, 0.1 and 0.2 were selected. It should be noted that the traffic-related HOA shows a remarkably similar spectral pattern as CCOA at $m/z < 120$ (Sun et al., 2016b), which cannot be separated with PMF-ACSM. Therefore, FFOA here represents a combined factor of traffic-related HOA and CCOA. In addition, the ME-2 results with a-value of 0.2 are also presented for comparisons.

Figure S2 shows a comparison of source apportionment results from three different approaches. While the monthly average

SOA (= LO-OOA + MO-OOA) and POA (= HOA + COA + BBOA) are highly correlated ($r^2 = 0.76$ and 0.89, respectively, Fig. S3), the seasonal ME2-ACSM reports an overall 16% higher SOA concentrations than the two-factor solution of PMF-ACSM. The largest differences occur mainly in cold season, for example November – March, which can be partially explained by the changes in LO-OOA. Comparatively, the contributions of POA and SOA are close in warm season, for example July – October. We also compared the results between the seasonal and entire dataset ME-2 analysis. As shown in

Fig. S2, the POA and SOA contributions present differences of 5 – 14% on average during the first eight months' measurements, while they are very consistent during the rest of months with the differences less than 3%. Figure S4 presents a comparison of POA and SOA contributions from ME2-ACSM in this study with those reported previously from PMF-ACSM (Sun et al., 2012;Jiang et al., 2013;Sun et al., 2013b;Sun et al., 2014;Jiang et al., 2015). The results are overall consistent except for winter 2011 – 2012, which shows a higher SOA contribution (54%) in this study than that (31%)

reported in Sun et al. (2013b). We found that such a difference was mainly caused by the contribution of LO-OOA (23.6%), which was not resolved in the Sun et al. (2013b) study with PMF-ACSM.

## 2.3 Potential source contribution function analysis

The potential source regions of five OA factors in each season were determined using potential source contribution function (PSCF) analysis (Polissar, 1999) with 72 hr back trajectories that were calculated hourly using the HYSPLIT model (Draxler

and Rolph, 2013) at a releasing height of 100 m. The back trajectories are counted in gridded cells (i,j), and the PSCF is calculated as the ratio of the number of points above a threshold value (75[th] percentile in this study, $m_{ij}$) to the total points ($n_{ij}$) in each grid cell. A weighing function that is the same as Sun et al. (2015) was further applied to the calculation to downweight cells associated with low values of $n_{ij}$. The regions with high PSCF values indicate the potential sources for high concentrations of OA factors.

## 3   Results and discussion



## 3.1 Mass concentrations, composition, and seasonal variations

Figure 2 shows the time series of organics, five OA factors and meteorological parameters for the entire study. OA presents dynamic variations during all seasons with hourly average concentrations ranging from 0.45 to 301 μg m$^{-3}$, and daily average values from 3.0 to 120 μg m$^{-3}$. The variations in OA are much larger in heating season than in summertime, mainly due to the more frequent changes between clean periods and polluted episodes (Sun et al., 2013b;Sun et al., 2015). Indeed, our previous studies showed a much higher frequency of clean periods (NR-PM$_1$ < 10 μg m$^{-3}$) in winter than summer (23% vs. 5%) (Sun et al., 2015). As shown in Fig. 3, OA presents a strong seasonal variation with monthly average concentrations ranging from 13.6 to 46.7 μg m$^{-3}$. The concentrations in wintertime are nearly twice those in summertime mainly due to the largely enhanced coal combustion emissions in heating season (Sun et al., 2015), and the highest concentration occurred in January 2013, a month with severe haze pollution (Sun et al., 2014;Wang et al., 2014). High OA concentration (monthly average of 37.2 μg m$^{-3}$) was also observed in June 2012, mainly due to the impacts of agricultural burning.

Five OA factors vary differently across different seasons. FFOA that is mainly associated with traffic and coal combustion emissions presents the strongest seasonal variation among the OA factors. The monthly average FFOA concentrations are 3.7 – 6.9 μg m$^{-3}$ and 4.4 – 8.5 μg m$^{-3}$ in the heating seasons of 2011-2012 and 2012-2013, respectively, which is much higher than 1.1 – 1.5 μg m$^{-3}$ in summer (Fig. 4a). Consistently, the contribution of FFOA to OA is significantly increased from 5 – 8% in summer to 13 – 22% in heating season (Fig. 3). The time series of FFOA (Fig. 2d) also shows a substantial increase after the heating season starts on 15 November in both 2011 and 2012, supporting the large impacts of coal combustion emissions on FFOA. Comparatively, FFOA in summer is expected to be mainly from traffic emissions considering that residential coal combustion emissions could not be significant. Assuming that traffic-related FFOA is relatively constant throughout the year, we then estimate an upper limit of ~70% of FFOA from coal combustion emissions in heating season (~ 30% from traffic emissions). This results is consistent with our previous high resolution aerosol mass spectrometer (HR-AMS) analysis in winter that HOA and CCOA on average contributed 10% and 20% to OA, respectively (Sun et al., 2016b).

The temporal variations of COA are characterized by pronounced daily peaks that are associated with cooking emissions (Fig. 2e). However, the seasonal difference of COA is much smaller than that of FFOA, for example, the COA concentration is 3.8 – 4.2 μg m$^{-3}$ in winter 2011 – 2012, which is only ~50 – 60% higher than that in summer 2011 (Fig. 4b). This is overall consistent with the facts that cooking emissions are expected to be relatively constant throughout the year. The seasonal differences in COA concentrations can be explained by the different mixing layer heights (MLH) in different seasons, for example, ~40% higher MLH during daytime in summer than winter (Tang et al., 2016). Despite the seasonal concentration differences, the contributions of COA to OA are relatively stable by varying from 10 to 15% except slightly higher values in April (17 – 18%). These results indicate that COA is an important source of OA during all seasons, consistent with previous results observed in Beijing (Huang et al., 2010;Elser et al., 2016;Hu et al., 2016). In fact, the contributions of COA to OA are



higher than those of FFOA by 3 – 10% in non-heating season, indicating that cooking emission is a more important primary source than traffic emissions in the megacity of Beijing.

BBOA shows a similarly pronounced seasonal variation as FFOA. As shown in Fig. 4c, the BBOA concentration increases gradually from summer ($< \sim 2$ μg m$^{-3}$) to winter (mostly $> 4$ μg m$^{-3}$) with an enhancement by a factor of more than 2. Consistently, the contribution of BBOA to OA shows an increase from 7 – 8% in summer to 11 – 14% in winter (Fig. 3). These results indicate that BBOA is a more important source of OA in winter than summer, which is overall consistent with the fact that biomass is also an important fuel for residential heating in northern China (Chen et al., 2017). However, we also observed high BBOA concentrations in June and October, two months with significant impacts from agricultural burning. Our subsequent measurements in June 2013 at a suburban site, Xianghe, which is approximately 50 km southeast of Beijing further support the significant agricultural burning impacts on OA, and the contribution of BBOA was increased from 11% to 21% during the periods with biomass burning (Sun et al., 2016c). However, compared with winter, the BB impacts in June and October are relatively short and usually last only a few days.

LO-OOA also presents a pronounced seasonal variation pattern, yet the highest concentration occurs in the heating season (Fig. 4d). This is not expected as the photochemical processing is more significant in summer. As indicated in Fig. 1, the mass spectral profiles of LO-OOA have much differences in different seasons in terms of $m/z$ 43/44 ratios and the fractions of large $m/z$'s. For example, the LO-OOA spectrum in winter presents much higher signals at $m/z > 60$, and the time series of LO-OOA is even correlated with primary aerosol species, e.g., FFOA ($r^2 = 0.58 - 0.75$), BC ($r^2 = 0.73$), and Chl ($r^2 = 0.58 - 0.65$) (Fig. S5). LO-OOA is also correlated with FFOA and Chl in spring and fall seasons, and high concentrations mainly occur during the periods with residential heating, i.e., 15 – 30 November and 1 – 15 March. These results suggest that LO-OOA in heating season is more like a combustion–related SOA that is formed under low temperature. Another possibility is that LO-OOA is still mixed with part of primary coal combustion OA, yet cannot be completely separated using the standard traffic-related HOA spectrum as a constrain. For example, previous HR-AMS studies in Beijing showed higher signals of large $m/z$'s in CCOA than HOA (Hu et al., 2016;Sun et al., 2016b). In comparison, LO-OOA is weakly correlated with other species in summer, supporting its different characteristics from those in winter. Note that LO-OOA shows high concentrations during the late June 2012 when there are significant biomass burning impacts, and the LO-OOA spectrum is remarkably similar to that of oxygenated OA influenced by biomass burning (OOA-BB) observed in Nanjing during harvest seasons (Zhang et al., 2015). These results suggest that a larger fraction of LO-OOA in June is likely from the oxidized BBOA. Indeed, a recent study of transported wildfire plumes showed that BBOA becomes significantly oxidized through atmospheric aging and that the mass spectra of aged BBOA can appear highly similar to low-volatility OOA (LV-OOA)(Zhou et al., 2017). In addition,  our previous studies at the suburban site (Xianghe) near Beijing also found that biomass burning aerosols can be rapidly oxidized to form a considerable fraction of LO-OOA (Sun et al., 2016c). LO-OOA



represents a large fraction of OA, on average accounting for 16 – 29% and 19 – 30% in heating seasons of 2011 – 2012 and 2012- 2013, respectively, and also as much as ~15 – 20% during other seasons except for the summer of 2012 (~10%).

The monthly average concentration of MO-OOA varies from 7.6 to 17.2 µg m$^{-3}$, and does not present a strong seasonal variation in the two years (Fig. 4e). Also, the highest concentrations of MO-OOA occur in different months in different years, for example, May – July in 2012 (15.2 – 17.4 µg m$^{-3}$) and January – March in 2013 (13.9 – 17.2 µg m$^{-3}$). MO-OOA is tightly correlated with secondary inorganic aerosol species including nitrate, sulfate, and ammonium during all seasons ($r^2$ > 0.60 for most of the time), and has slightly better correlations with nitrate particularly in winter (Fig. S5). Although the MO-OOA spectrum is similar during all seasons and also resembles that of low volatility OOA (LV-OOA) (Ng et al., 2011a), MO-OOA is still likely a mixed factor with different types of SOA. It cannot be separated with ME-2 in this study due to the limited chemical resolution of the ACSM measurements, for instance, the aqueous-phase SOA factor observed in winter (Sun et al., 2016b). Our results appear to be different from previous findings that LO-OOA tends to represent semi-volatile SOA that is generally correlated with nitrate, while MO-OOA usually represents low-volatility OOA that is generally correlated with sulfate (Lanz et al., 2007;Zhang et al., 2011). In fact, previous studies in Beijing show that LO-OOA is often weakly correlated with nitrate, for example, $r^2$ < 0.12 in fall 2015 (Zhao et al., 2017), likely indicating the different chemical processing between LO-OOA and nitrate. For example, the nighttime heterogeneous formation of nitrate was recently found to be similarly important as that of gas-particle partitioning of nitric acid (Wang et al., 2017), while the nighttime formation of LO-OOA is not clear yet. In addition, the similarly tight correlations between MO-OOA and secondary inorganic species highlight another fact that secondary aerosols in Beijing can have large contributions from regional transport (Sun et al., 2014). This is also consistent with the simultaneous increases during the formation stage of most polluted episodes (Zhao et al., 2013;Sun et al., 2016a). Overall, MO-OOA constitutes the largest fraction of OA mass among five OA factors during all seasons, and the contributions present a pronounced seasonal variation with the highest values in summer (47 – 64%) and the lowest values in winter (30 – 34%).

Table 1 presents a summary of monthly average concentrations and fractions of five OA factors. It is clear that the contributions of POA and SOA show strong seasonal differences. SOA (= LO-OOA + MO-OOA) shows the largest contribution to OA during the non-heating season by accounting for 65 – 75%. Although the contributions decrease to 52 – 64% in heating season, they are still higher than those of POA, indicating that SOA plays a more important role controlling OA chemistry than POA during all seasons. Consistently, the dominance of SOA in OA mass has been widely reported at various urban sites (Zhang et al., 2007;Jimenez et al., 2009). We also noticed the seasonal changes in SOA composition. For example, the contributions of LO-OOA generally exceed those of MO-OOA in heating season. Compared with SOA, POA (= FFOA + BBOA + COA) shows a large increase from ~30% in summer to ~40 – 50% in winter, and such increases are mainly driven by FFOA and BBOA from coal combustion and biomass burning emissions, respectively, supporting the large impacts of these two sources on POA in heating season.





## 3.2 Chemically-resolved OA composition

Figure 5 shows chemically-resolved OA composition as a function of OA mass loadings during four seasons. In summer, the SOA contribution first shows an increase at low OA mass loadings (< 20 μg m$^{-3}$), and then remains at relatively high values (~70 – 74%) at high mass loadings. These results indicate that SOA plays a dominant role in OA across different OA mass loadings in summer. However, we observe a significant change in SOA composition as a function of OA mass loadings. In particular, LO-OOA shows large increases from ~10% to nearly 30% associated with corresponding decreases in MO-OOA as OA mass loadings increase to > 40 μg m$^{-3}$. These results suggest that the periods with high OA loadings facilitate the formation of less oxidized SOA. A change in POA composition was also observed in summer 2011, which is characterized by an increase in FFOA and a corresponding decrease in COA as a function of OA mass loadings. As discussed in Section 3.3, FFOA in summer presents a similar pronounced diurnal profile as COA suggesting that FFOA and COA might not be well separated even with the constrained mass spectral profiles. In fact, the contributions of the sum of FFOA and COA are relatively stable across different OA mass loadings, which are ~20 – 25% except the period of OA > 65 μg m$^{-3}$ in the summer of 2012 with significant BB impacts.

Similar mass loading dependent OA composition is also observed in the fall season, yet with some differences between 2011 and 2012 (Fig. 5b). The SOA contribution in fall 2011 first shows a large increase from ~50% to 65% at OA < 30 μg m$^{-3}$, then remains at relatively constant levels (67 – 69%) at high OA mass loadings. In comparison, it is consistently high (65 – 70%) at OA < 50 μg m$^{-3}$ with a slight decrease to ~60% as OA mass loading increases in the fall of 2012. We also noticed large differences in BBOA and LO-OOA contributions at low mass loadings (OA < 15 μg m$^{-3}$) between 2011 and 2012, yet the total contribution of BBOA and LO-OOA was close. One of the reasons is due to the uncertainties in splitting BBOA and LO-OOA with the ME-2 analysis. Indeed, the mass spectrum of LO-OOA resembles to that of BBOA during the two fall seasons (Fig. 1). Although the contributions of the sum of FFOA and COA are relatively constant across different mass loadings, which are 21 – 25% and 23 – 27% in the fall of 2011 and 2012, respectively, we observed a clear increase in FFOA contribution by nearly a factor of 2 as OA mass loading increases. One reason is due to the half month period of the fall season (15 – 30 November) when FFOA shows a large increase because of residential heating (Wang et al., 2015). This is particularly clear for the bins with the largest OA mass loading (80 – 85 and 90 – 95 μg m$^{-3}$ in F11 and F12, respectively), which occurred mainly during the half month period of the heating season (Fig. 2). As a result, the contribution of FFOA increased to 23% and 27% in 2011 and 2012, respectively, which was associated with a large decrease in MO-OOA (15 and 22%, respectively).

The POA composition changes significantly as a function of OA mass loading in winter (Fig. 5c). The contribution of FFOA gradually increases from 3 – 7% to 22 – 28% when OA mass loading increases from less than 20 μg m$^{-3}$ to > 60 μg m$^{-3}$, highlighting a largely enhanced role of FFOA during highly polluted periods. In comparison, the COA contribution shows a



large decrease from ~30% to < 10% although the mass concentration is relatively stable at ~ 5-6 µg m⁻³, consistent with our previous conclusion that COA is a more important POA during clean periods with low OA mass loadings. The total contribution of SOA shows a small decrease by 5 – 10% at high OA mass loading periods mainly due to the decreases in MO-OOA, yet it is still as high as 52 – 58%. Such a result indicates that SOA is still important during the highly polluted

periods in winter (Huang et al., 2014). The changes in OA composition as a function of OA mass loading in spring are similar to those in fall. The SOA contribution shows a decrease during more polluted periods, yet it is still higher than that of POA (56 – 64% vs. 36 – 44%). Again, FFOA shows a large increase at high OA mass loading. For example, the FFOA contribution increases from 6 – 9% to ~20% as OA increases from < 20 µg m⁻³ to > 50 µg m⁻³. This is consistent with the fact that the spring season is also consisted of half month of heating season when OA presents the highest mass loading (Fig.

10    2)

### 3.3  Diurnal variations of OA factors

The diurnal profiles of five OA factors in four seasons are shown in Fig. 6. Similar to previous studies (Sun et al., 2011b;Ge et al., 2012), COA presents similar and pronounced diurnal cycles during all seasons, which are characterized by two peaks during lunch and dinner times respectively. The nighttime COA peak is nearly twice that of noontime during all seasons

except for summer indicating higher cooking emissions at nighttime coupled with shallower boundary layer height compared to the middle of the day. The contribution of COA to OA is correspondingly high at nighttime (~20 – 25%), supporting the significant impact of cooking activities on OA loading in urban areas. FFOA presents the lowest concentrations throughout the day in summer, yet two small peaks corresponding to mealtimes are also observed. Such a diurnal profile is not expected as the traffic-related HOA typically peaks during rush hours. One of the reasons is that the FFOA factor was not fully

resolved and was partially influenced by cooking organic aerosol. Previous summer studies (Sun et al., 2010;Sun et al., 2012;Zhang et al., 2015) using quadrupole AMS or ACSM have indeed shown PMF analysis has difficulty resolving traffic-related HOA from COA in summer due to their relatively similar mass spectra measured by unit mass resolution instruments. In comparison, PMF analysis of the high resolution mass spectra of OA is able to separate HOA from COA in summer in Beijing (Huang et al., 2010;Hu et al., 2016), and the results show substantially different diurnal profiles between

these two factors. Compared to summer, FFOA and COA can be relatively well separated during other seasons as indicated by the much smaller noon peaks of FFOA. As shown in Fig. 6, FFOA presents much higher concentrations at nighttime than daytime due to 1) the enhanced coal combustion emissions, and 2) traffic emissions from diesel trucks and heavy duty vehicles that are only allowed inside the city between 22:00 – 6:00 (Han et al., 2009), and 3) shallow boundary layer. We also found that COA exceeds FFOA before midnight during most of the seasons although it shows a rapid decrease after

21:00 when cooking activities are significantly reduced. FFOA then becomes a more important primary OA after midnight until ~11:00.  These results indicate the different roles of primary OA in different time of the day. Compared to FFOA and COA, the diurnal profiles of BBOA are less pronounced with slightly higher concentrations at nighttime.



The diurnal profiles of the two SOA factors are different during most of the seasons. As shown in Fig. 6c, MO-OOA presents the largest increase during daytime in winter, and the concentration is continuously increased by a factor of 2 from 8:00 until 19:00, indicating the strong photochemical production of SOA. The continuous daytime increase of MO-OOA is interrupted by a temporal decrease in the late afternoon during the other three seasons, which is very likely due to the rising

boundary layer height with the dilution effect exceeding the secondary production. Compared to the diurnal changes in mass concentrations, the contribution of MO-OOA to OA shows similar diurnal trends during all seasons (Fig. S6), and the contributions increase gradually from the early morning and reach the maximum values at 16:00 – 17:00. For example, the contributions increase from 30% to 48% and 53% in winter 2011 and 2012, respectively (Fig. S6c), and from ~40% to ~56% in the fall seasons (Fig. S5b). Although the contributions are decreased slightly at lunch time in summer due to the largely

increased COA, they are still as high as 47 – 52%. Also note that MO-OOA constitutes the largest fraction in OA throughout the day during all seasons, highlighting the ubiquity and dominance of SOA at urban sites (Zhang et al., 2007;Jimenez et al., 2009).

The diurnal profiles of LO-OOA are overall similar during three seasons – winter, spring and autumn. The concentrations decrease gradually during daytime, reach the minimum values typically between 16:00 – 17:00, and then increase rapidly

during the rest time of the day. Such diurnal profiles are exactly opposite to those of temperatures, likely indicating that gas-partitioning driven by $T$ plays the dominant role. Higher concentrations at nighttime than daytime also resemble somewhat to those of FFOA, which highlights another possibility that LO-OOA is combustion-related SOA as discussed above. It should be noted that the decreases in LO-OOA correspond to the increases in MO-OOA, which likely indicates the continuous transformation of LO-OOA into MO-OOA due to photochemical aging (Morgan et al., 2010;Sun et al., 2011a). LO-OOA

presents a very different diurnal profile in summer compared to those during the other three seasons, which is characterized by similar COA peaks at lunch and dinner times. One explanation is that LO-OOA was not well separated from COA in summer and was mixed with part of COA. Another explanation is that part of LO-OOA in summer was from the oxidation of VOCs from cooking emissions, which is consistent with a recent study showing that aging of different cooking aerosol can form less oxidized SOA (O/C = 0.24 – 0.46) (Liu et al., 2017b). Overall, SOA dominates OA throughout the day during all

seasons. The contributions vary between 60 – 80% in spring, summer and fall although they are decreased at ~50 – 60% in winter. These results indicate that secondary formation processes in winter via either photochemical and/or aqueous-phase reactions could be similarly important as primary emissions.

### 3.4 RH/T dependence of OA

The RH/$T$ dependence of NR-PM$_1$ species in Beijing is presented in Sun et al. (2015). Here, we mainly focus on the impacts

of RH and $T$ on OA factors. As shown in Fig. 7a1, FFOA presents the highest concentrations in the right-bottom region with low $T$ and high RH. This is consistent with the significantly enhanced primary emissions (e.g., coal combustion and biomass





burning) in winter. At similarly low $T$ levels, FFOA concentrations increase substantially as a function of RH particularly at RH > 50%. In winter, the periods with high RH levels are typically characterized by stagnant meteorological conditions, e.g., low wind speed (Fig. S7a) and shallow boundary layer height (Sun et al., 2015), which lead to the accumulation of primary pollutants and the increases in FFOA concentrations. However, the ratios of FFOA/OA also present high values in the

similar right-bottom region (Fig. 7a2) indicating that high RH conditions could also facilitate the formation of more FFOA from coal combustion emissions (Sun et al., 2013a).

Compared with FFOA, the concentration gradients of COA as a function of RH/$T$ are much smaller (Fig. 7b). The COA emissions are relatively constant throughout the year and are not expected to have a strong RH/$T$ dependence. Thus, the concentration gradients of COA would be mainly caused by the dilution or accumulation effects. For example, low

concentrations in the left region (RH < 30%) are mainly caused by high WS (Fig. S7a), while high values in the right-bottom region are mainly due to the stagnant conditions which is consistent with those of FFOA. However, the RH/$T$ dependence of COA/OA is largely different from that of FFOA/OA. In particular, COA/OA presents the highest values at low RH levels (<30%) across different $T$ levels. These results highlight the dominant role of COA in OA in clean periods during all seasons. We further evaluated the RH/$T$ dependence of FFOA/COA (Fig. S7b). Because FFOA is from both traffic and

seasonal dependent coal combustion emissions, the ratio of FFOA/COA can be used as a diagnostic for the impacts of source emissions. Indeed, the highest FFOA/COA ratio (> 1.5) occurred when $T$ < 6 $^o$C and RH > 40% (Fig. S7b), supporting the facts that 1) more FFOA emissions, e.g., from coal combustion, at low $T$, and 2) more FFOA formation at high RH levels.

The RH/$T$ dependence of BBOA is similar to that of FFOA (Fig. 7c), which shows higher concentrations in right-bottom region, indicating more biomass burning emissions at high RH levels in winter. The contribution of BBOA to OA is

relatively constant at ~6 – 10% across different RH and $T$ levels except for the high contribution in the region with RH > 70% and low $T$ (-6 – 0 $^o$C). Such a result could be potentially important for a better understanding of aqueous-phase SOA formation as a recent study found that aqueous-phase processing of biomass-burning emissions can contribute to ambient SOA formation substantially (Gilardoni et al., 2016).

The RH/$T$ dependence of two SOA factors are very different. As shown in Fig. 7d, LO-OOA presents the highest

concentrations in the region with RH > 40% and T < 6$^o$C. Although such a distribution is similar to that of FFOA, we also observed moderately high concentrations at high $T$ when RH is above 60%. The RH/T dependence of LO-OOA/OA is also different from those of other OA factors. The dominance of LO-OOA (~30%) is observed in the bottom region with $T$ below 0 $^o$C, while it was similarly important in other regions with different RH and $T$ levels. These results indicate two different formation mechanisms of LO-OOA. While gas-particle partitioning plays a dominant role in LO-OOA formation at low $T$,

photochemical production becomes more important during other periods. Compared with LO-OOA, the MO-OOA concentrations show strong RH dependence during all seasons while the $T$ dependence is not clear (Fig. 7e). Such a



distribution illustrates that aqueous-phase processing might be an important mechanism in the formation of MO-OOA. This is also supported by the correlations between MO-OOA and sulfate, a species dominantly from cloud and aqueous-phase processing, during all seasons. However, the contribution of MO-OOA to OA presents a strong $T$ dependence with much higher contributions at $T > 18^{\circ}C$ than those below $12^{\circ}C$ ($\sim > 50\%$ vs. $\sim 30 – 40\%$). Such a $T$ dependent MO-OOA/OA clearly

highlights the importance of MO-OOA in summer than other seasons across different RH levels.

## 3.5  Potential source regions of OA

Figure 8 shows the PSCF analysis of five OA factors during four seasons. FFOA shows high PSCF values in a small region near the sampling site in the summer seasons of 2011 and 2012, supporting the dominant source of local emissions. This is consistent with the fact that FFOA in summer is mainly from traffic-related emissions. While FFOA shows a similarly

dominant local source in the fall of 2011, it is also characterized by a high PSCF region to the southwest of Beijing in the fall of 2012. These results indicate that FFOA can have very different contributions from local sources and regional transport in the same season in different years. In the winters of 2011 and 2012, FFOA shows high PSCF values in the regions located to the south and southwest of Beijing, indicating that regional transport plays an important role for the high concentrations of FFOA in winter in Beijing. In fact, the coal boilers for residential heating have been all replaced with natural gas inside the

Beijing city, and a large contribution of FFOA at the sampling site would be expected from regional transport (Cheng et al., 2016;Sun et al., 2017). For example, Sun et al. (2014) found that regional transport can contribute 84% of coal combustion emissions during the peak of severe haze pollution in 12-13 January 2013. Note that the high PSCF is not beyond the cities of Shijiazhuang and Hengshui, suggesting that the transport is mainly dominated in a region less than 200 km. This is consistent with the results from a recent study that the regional transport during severe haze episodes is dominantly from the

south/southwest (e.g., Baoding, Cangzhou, and Hengshui) and south/southeast (e.g., Langfang and Tianjin), while that from farther regions, e.g., Shijiazhuang, Xiangtai, and Handan are negligible (Li et al., 2017a). The contribution of regional transport to high FFOA concentrations are even more important in spring seasons of 2012 and 2012, which is characterized by a narrow high PSCF band along Mountain Taihang. The cities of Baoding and Hengshui (approximately 250 km from Beijing) are both potential source regions of Beijing FFOA. These results are consistent with the wind rose plots in Fig. S8

showing more frequency of high WS from the southwest in spring than winter. Therefore, reducing coal combustion emissions in the regions to the south/southwest of Beijing would greatly help mitigate FFOA pollution in Beijing in winter season.

Compared to FFOA, the regions with high PSCF values for COA are much smaller during all seasons, consistent with the fact that COA is dominantly from local cooking emissions. A recent modeling study of COA in UK also found that the COA

concentrations decrease rapidly outside of the city and could not have a significant impact on rural areas (Ots et al., 2016). However, a small high PSCF region (typically less than 50 km) located to the east/southeast is frequently observed in many





seasons, indicating that a short-distance transport could contribute the high COA at our sampling site. As shown in Fig. 8c, the PSCF regions of BBOA are very similar to those of FFOA in fall, winter, and spring seasons. We also noticed that BBOA is moderately correlated with FFOA during these three seasons ($r^2 = 0.44 - 0.80$, Fig. S5), suggesting similar seasonal emission characteristics between BBOA and FFOA. Previous studies have shown that residential emissions

contribute substantially to the regional air pollution in northern China in winter (Liu et al., 2016). While coal is the dominant fuel for residential heating in winter, biomass is also an important fuel (Liu et al., 2017a). Not surprising, coal combustion and biomass burning show similar temporal variations in winter. As we discussed above, high concentrations of FFOA and BBOA in spring and fall mainly occur during the half month period in heating season (15 – 30 November and 1 – 15 March), and the high PSCF regions of BBOA and FFOA in these two seasons are consistently contributed by the high concentrations

during these two periods. The PSCF region of BBOA is quite different from that of FFOA in summer, indicating their different source regions. While FFOA in summer is dominantly contributed by local traffic emissions, BBOA still has an important source from regional transport.

The potential source regions of the two SOA factors are quite different in different seasons (Figs. 8d and 8e). In the spring seasons of 2012 and 2013, both LO-OOA and MO-OOA show similar potential source regions as those of FFOA and BBOA

with high PSCF values located to the southwest of Beijing. These results indicate that regional transport from the southwest is an important contribution of both primary and secondary OA. Figure 8e also shows nearly twice farther high potential source region of MO-OOA than LO-OOA and FFOA in spring 2012, consistent with the formation of more oxidized SOA over a wider regional scale. During the other three seasons, MO-OOA shows ubiquitously wider potential source regions than LO-OOA, suggesting that the formation of MO-OOA is more regional than LO-OOA. This is also consistent with the

good correlations between MO-OOA with sulfate that is mainly formed via regional cloud processing. For instance, the potential source region of LO-OOA is limited in a small region to the south of Beijing in summer 2011, while that of MO-OOA is beyond approximately 250 km. These results suggest that LO-OOA in summer is likely mainly from local photochemical production while MO-OOA has more contributions from regional transport. Even in the same season, the source regions could be different. For example, LO-OOA presents two potential source regions located to the southwest and

southeast of Beijing, respectively, while that of MO-OOA is dominantly from the southeast. Note that a potential source region over the Bohai sea was also observed for MO-OOA. One explanation is that MO-OOA from a regional scale was circulated from the Bohai sea before arriving at Beijing. Compared to other seasons, the potential source region of MO-OOA in winter is much smaller, which is mainly located to the southwest in 2011, and the whole south of Beijing in winter 2012, while the highly polluted cities, e.g., Shijiazhuang and Hengshui in Hebei province, appear not to contribute significantly to

both POA and SOA in Beijing. These results highlight the importance of employing different control strategies in reducing the number of severely polluted days in Beijing in different seasons, in particular, more efforts are needed to control emissions in the surrounding regions within 200 km in winter.



## 4    Conclusions

An aerosol chemical speciation monitor was deployed at an urban site in Beijing for two years' real-time measurements of NR-PM$_1$ species from July 2011 to May 2013. Organics present a pronounced seasonal variation with the average concentration in winter being twice higher than the average in summer mainly due to coal combustion emissions in the heating season. The ME-2 analysis of seasonal datasets identified five OA factors during all seasons, including FFOA from traffic and coal combustion emissions, BBOA, COA, LO-OOA and MO-OOA. FFOA and BBOA present similar seasonal variations with high concentrations in heating season, which is mainly caused by enhanced coal combustion and biomass burning emissions in winter. Another support is the pronounced diurnal cycles of the two factors with much higher concentrations at nighttime due to the increased emissions for residential heating. Comparatively, the seasonal variation of COA is much less pronounced, and the contributions are relatively constant (10 – 15%) across different seasons. High concentration of BBOA is also observed in June due to agricultural burning impacts. The seasonal variation of LO-OOA is more similar to that of FFOA and BBOA, likely indicating that LO-OOA is a combustion-related SOA, and gas-particle partitioning under low $T$ plays a major role. This is also consistent with the diurnal profile which presents much higher concentrations at nighttime than daytime during most of the seasons. MO-OOA constitutes the largest fraction in OA during all seasons, and the contributions present a pronounced seasonal variation with the highest values in summer (47 – 64%) and the lowest values in winter (30 – 34%).

The chemically-resolved OA composition demonstrates the different roles of OA factors in different levels of pollution. While SOA dominates OA across different mass loadings during all seasons, a large increase in FFOA contribution as a function of OA mass loading was also observed particularly in winter indicating an enhanced role of coal combustion emissions during severely polluted episodes. We also noticed a change in SOA composition as the increase of OA mass loading, which is characterized by an increase in LO-OOA associated with a corresponding decrease in MO-OOA. Such a result likely suggests an influence of OA mass loadings on aging of different SOA factors (Shilling et al., 2009). The RH/$T$ dependence of OA factors shows ubiquitously high concentrations at low $T$ and high RH, elucidating the most frequency of severe haze episodes at high RH conditions in winter. COA presents the weakest RH/$T$ dependence among OA factors, yet the contribution to OA is much higher at RH < 30%, highlighting the important role of COA during clean periods. MO-OOA shows higher concentrations at higher RH levels across different $T$ levels, and also correlated with both nitrate and sulfate during all seasons, indicating the formation of MO-OOA from both aqueous-phase and photochemical processing. The potential source regions of five OA factors are investigated with PSCF analysis. FFOA shows high potential source regions located to the south/southwest of Beijing during three seasons except summer with dominant sources from local emissions. These results indicate that regional transport is also important for high FFOA concentrations in heating season in Beijing in winter. Compared with FFOA, COA is dominantly from local emissions and the small regions near the sampling site. The



two SOA factors show quite different potential source regions. In particular, MO-OOA shows much wider source regions than the other OA factors, supporting the formation of highly oxidized SOA over a regional scale.

## 5    Abbreviations

| | |
|---|---|
| ACSM | Aerosol Chemical Speciation Monitor |
| HR-AMS | High-resolution Time-of-Flight Aerosol Mass Spectrometer |
| CAPS | Cavity Attenuated Phase Shift |
| NR-PM$_1$ | Non-refractory Submicron Aerosol |
| OA | Organic Aerosol |
| SOA | Secondary Organic Aerosol |
| POA | Primary Organic Aerosol |
| FFOA | Fossil Fuel Related Organic Aerosol |
| HOA | Hydrocarbon-like Organic Aerosol |
| COA | Cooking Organic Aerosol |
| CCOA | Coal Combustion Organic Aerosol |
| BBOA | Biomass Burning Organic Aerosol |
| OOA | Oxygenated Organic Aerosol |
| LO-OOA | Less-oxidized Oxygenated Organic Aerosol |
| MO-OOA | More-oxidized Oxygenated Organic Aerosol |
| LV-OOA | Low Volatility Oxygenated Organic Aerosol |
| ME-2 | Multilinear Engine |
| PMF | Positive Matrix Factorization |
| PSCF | Potential Source Contribution Function |
| VOCs | Volatile Organic Compounds |
| WD | Wind Direction |
| WS | Wind Speed |
| RH | Relative Humidity |
| $T$ | Temperature |
| MLH | Mixing Layer Height |

## 5    Acknowledgments

This work was supported by the National Natural Science Foundation of China (41575120, 91744207, 41571130034) and the National Key Project of Basic Research (2014CB447900).



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





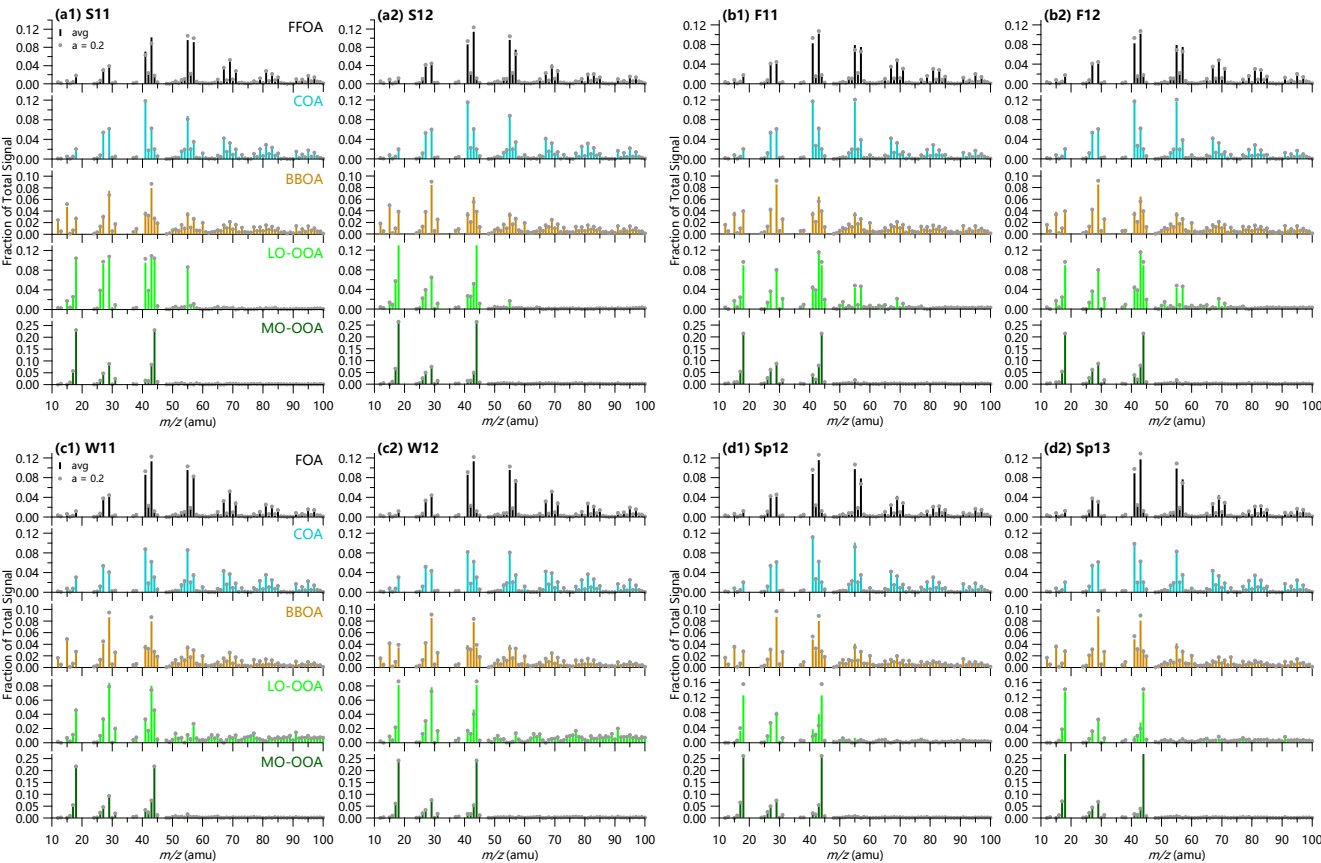

**Figure 1:** Mass spectral profiles of five OA factors from ME2-ACSM analysis in (a1) S11, summer 2011, (a2) S12, summer 2012, (b1) F11, fall 2011, (b2) F12, fall 2012, (c1) W11, winter 2011, (c2) W12, winter 2012, (d1) Sp12, spring 2012, and (d2) Sp13, spring 2013.







**Figure 2:** Time series of meteorological parameters (a) WS and (b) RH and *T*, and hourly average mass concentrations of (c) organics (Org), and (d-h) five OA factors, i.e., FFOA, COA, BBOA, LO-OOA and MO-OOA. The period of heating season (15 November – 15 March) is also marked as shaded areas.



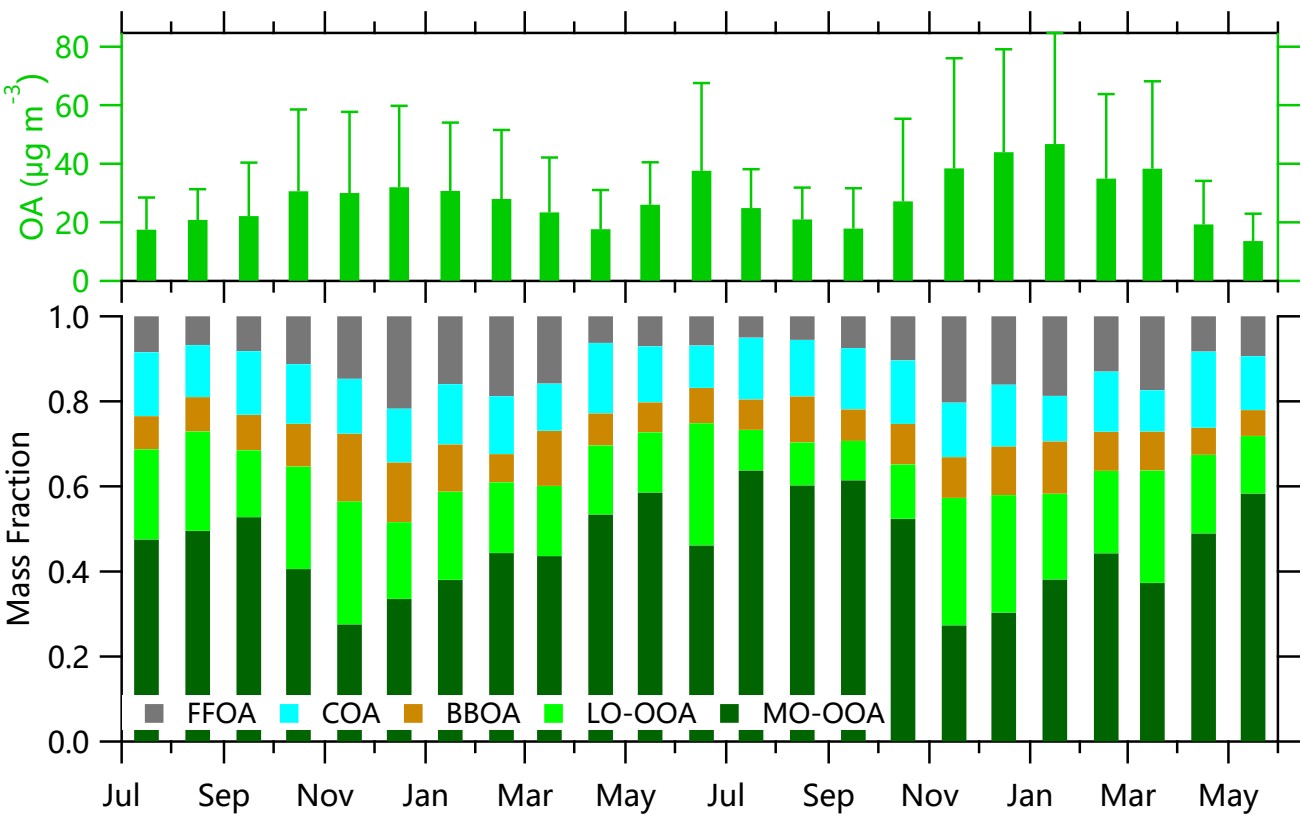

**Figure 3:** Monthly average OA mass concentrations and composition. The error bars represent one standard deviations of monthly averages.



**Figure 4:** Monthly average mass concentrations of five OA factors. The bars are the average of ME-2 results from three model runs, i.e., a = 0, 0.1, and 0.2. The results with a = 0.2 are also shown for a comparison. The error bars represent one standard deviations of monthly averages.



**Figure 5:** Variations of OA composition as a function of OA mass loadings during four seasons. The data are grouped in OA bins (5 µg m$^{-3}$ increment). The white circles indicate the frequency of each OA bin and the total number of points (*N*) for each season is also shown.

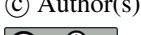



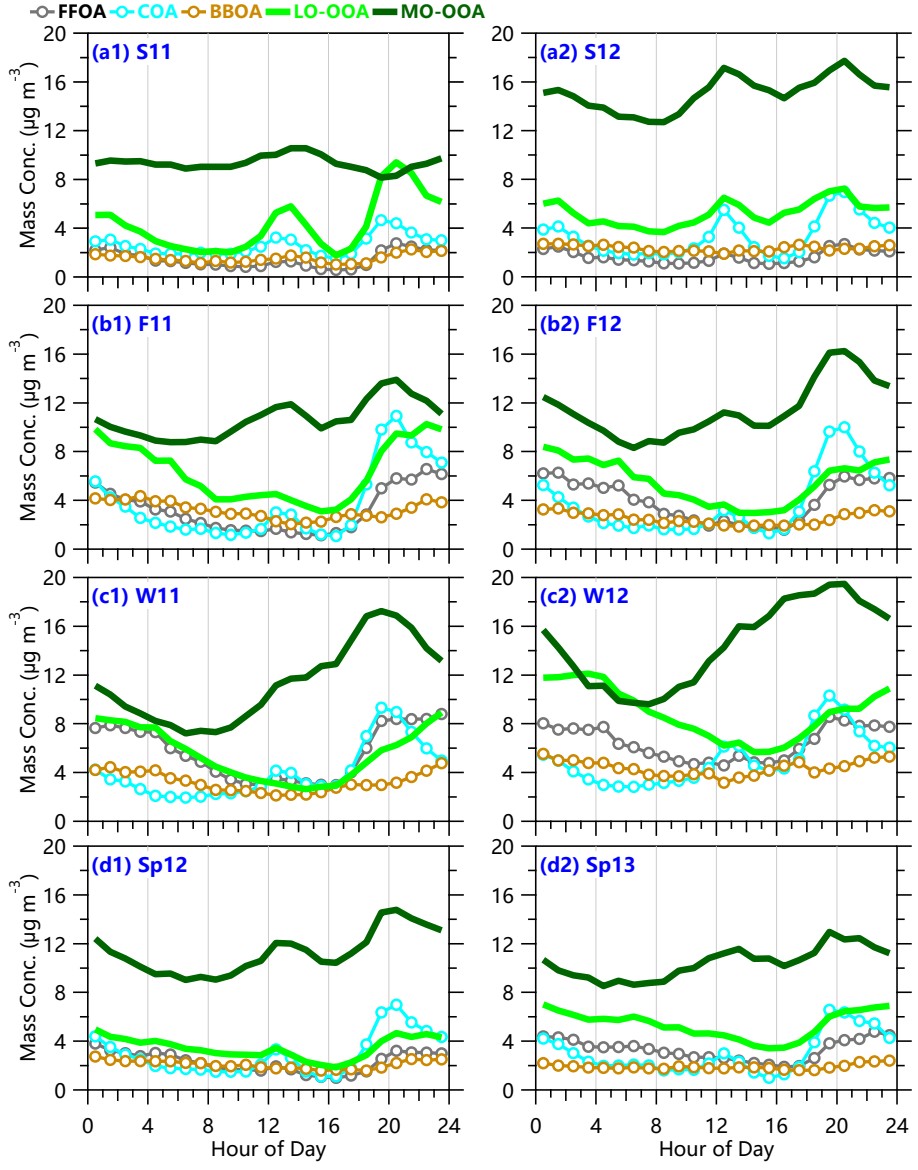

**Figure 6:** Average diurnal cycles of five OA factors during four seasons.





**Figure 7.** RH/$T$ dependence of mass concentrations and mass fractions of five OA factors for the entire study. The data are grouped into grids with increments of RH and $T$ being 5% and 3 °C, respectively. Grid cells with the number of data points fewer than 10 are excluded.



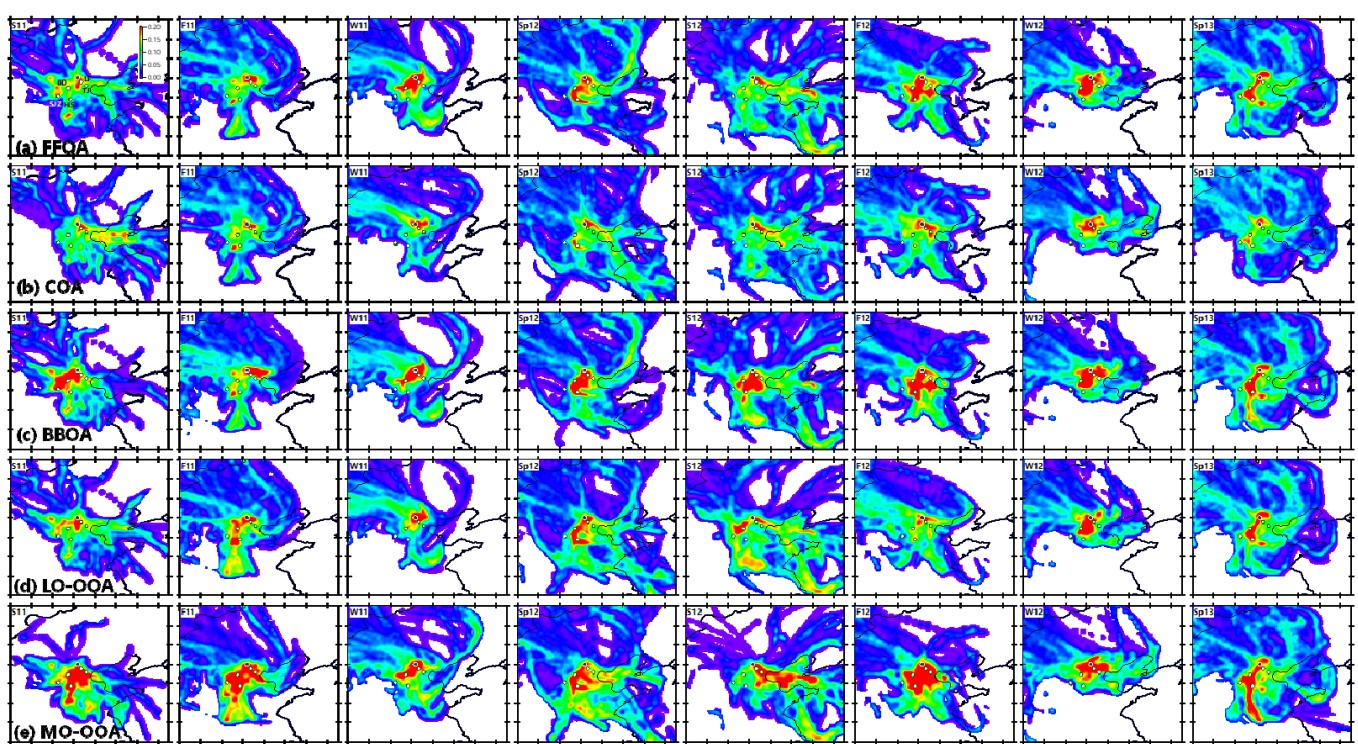

**Figure 8:** PSCF of five OA factors in each season. The cities marked as solid circles in each panel are Tianjing (TJ), Langfang (LF), Baoding (BD), Shijiazhuang (SJZ), and Hengshui (HS). The color scales indicate the values of PSCF.



**Table 1:** A summary of monthly average mass concentrations and mass fractions of five OA factors, POA (= FFOA+ COA+ BBOA) and SOA (= LO-OOA + MO-OOA).

| Month-Year | Mass Concentrations ($\mu g\ m^{-3}$) | | | | | | | Mass Fractions of OA | | | | | | |
|---|---|---|---|---|---|---|---|---|---|---|---|---|---|---|
| | FFOA | COA | BBOA | LO-OOA | MO-OOA | POA | SOA | FFOA | COA | BBOA | LO-OOA | MO-OOA | POA | SOA |
| Jul-11 | 1.5 | 2.6 | 1.4 | 3.7 | 8.4 | 5.5 | 12.1 | 0.08 | 0.15 | 0.08 | 0.21 | 0.47 | 0.31 | 0.69 |
| Aug-11 | 1.4 | 2.6 | 1.7 | 4.9 | 10.4 | 5.7 | 15.3 | 0.07 | 0.12 | 0.08 | 0.23 | 0.49 | 0.27 | 0.73 |
| Sep-11 | 1.8 | 3.3 | 1.8 | 3.5 | 11.7 | 7.0 | 15.2 | 0.08 | 0.15 | 0.08 | 0.16 | 0.53 | 0.31 | 0.69 |
| Oct-11 | 3.4 | 4.3 | 3.1 | 7.3 | 12.3 | 10.8 | 19.7 | 0.11 | 0.14 | 0.10 | 0.24 | 0.41 | 0.35 | 0.65 |
| Nov-11 | 4.4 | 3.8 | 4.7 | 8.6 | 8.2 | 12.9 | 16.7 | 0.15 | 0.13 | 0.16 | 0.29 | 0.28 | 0.44 | 0.56 |
| Dec-11 | 6.9 | 4.0 | 4.4 | 5.7 | 10.6 | 15.3 | 16.3 | 0.22 | 0.13 | 0.14 | 0.18 | 0.33 | 0.48 | 0.52 |
| Jan-12 | 4.7 | 4.2 | 3.3 | 6.2 | 11.3 | 12.3 | 17.5 | 0.16 | 0.14 | 0.11 | 0.21 | 0.38 | 0.41 | 0.59 |
| Feb-12 | 5.2 | 3.8 | 1.8 | 4.6 | 12.3 | 10.8 | 16.9 | 0.19 | 0.14 | 0.07 | 0.17 | 0.44 | 0.39 | 0.61 |
| Mar-12 | 3.7 | 2.6 | 3.0 | 3.9 | 10.3 | 9.4 | 14.1 | 0.16 | 0.11 | 0.13 | 0.16 | 0.44 | 0.40 | 0.60 |
| Apr-12 | 1.1 | 2.9 | 1.3 | 2.8 | 9.2 | 5.3 | 12.1 | 0.06 | 0.17 | 0.08 | 0.16 | 0.53 | 0.30 | 0.70 |
| May-12 | 1.8 | 3.4 | 1.8 | 3.7 | 15.2 | 7.1 | 18.9 | 0.07 | 0.13 | 0.07 | 0.14 | 0.59 | 0.27 | 0.73 |
| Jun-12 | 2.6 | 3.8 | 3.1 | 10.9 | 17.4 | 9.5 | 28.2 | 0.07 | 0.10 | 0.08 | 0.29 | 0.46 | 0.25 | 0.75 |
| Jul-12 | 1.2 | 3.5 | 1.7 | 2.3 | 15.2 | 6.4 | 17.5 | 0.05 | 0.15 | 0.07 | 0.10 | 0.64 | 0.27 | 0.73 |
| Aug-12 | 1.1 | 2.7 | 2.2 | 2.0 | 12.1 | 6.0 | 14.2 | 0.06 | 0.13 | 0.11 | 0.10 | 0.60 | 0.30 | 0.70 |
| Sep-12 | 1.3 | 2.5 | 1.3 | 1.6 | 10.7 | 5.1 | 12.3 | 0.08 | 0.14 | 0.07 | 0.09 | 0.61 | 0.29 | 0.71 |
| Oct-12 | 2.7 | 3.8 | 2.4 | 3.3 | 13.4 | 8.9 | 16.6 | 0.10 | 0.15 | 0.10 | 0.13 | 0.52 | 0.35 | 0.65 |
| Nov-12 | 7.6 | 4.8 | 3.6 | 11.2 | 10.2 | 16.0 | 21.4 | 0.20 | 0.13 | 0.10 | 0.30 | 0.27 | 0.43 | 0.57 |
| Dec-12 | 6.3 | 5.7 | 4.4 | 10.8 | 11.8 | 16.4 | 22.6 | 0.16 | 0.15 | 0.11 | 0.28 | 0.30 | 0.42 | 0.58 |
| Jan-13 | 8.5 | 4.8 | 5.6 | 9.2 | 17.2 | 18.9 | 26.4 | 0.19 | 0.11 | 0.12 | 0.20 | 0.38 | 0.42 | 0.58 |
| Feb-13 | 4.4 | 4.7 | 3.1 | 6.5 | 14.8 | 12.1 | 21.2 | 0.13 | 0.14 | 0.09 | 0.19 | 0.44 | 0.36 | 0.64 |
| Mar-13 | 6.5 | 3.6 | 3.4 | 9.9 | 13.9 | 13.5 | 23.7 | 0.17 | 0.10 | 0.09 | 0.26 | 0.37 | 0.36 | 0.64 |
| Apr-13 | 1.5 | 3.3 | 1.2 | 3.4 | 9.0 | 6.0 | 12.5 | 0.08 | 0.18 | 0.06 | 0.18 | 0.49 | 0.33 | 0.67 |
| May-13 | 1.2 | 1.6 | 0.8 | 1.8 | 7.6 | 3.7 | 9.3 | 0.09 | 0.13 | 0.06 | 0.14 | 0.58 | 0.28 | 0.72 |

