# Peer review of "Source apportionment of organic aerosol from two-year highly timeresolved measurements by an aerosol chemical speciation monitor in Beijing, China"

_Atmospheric Chemistry and Physics, 2017_

## Referee Comment (RC1) · Anonymous Referee #1 · 3 Mar 2018

Aerosol source apportionment is crucial for employing effective control strategies to improve air quality. This study presents seasonal and diurnal variations of different OA factors based on long-term (2 years) observations, and analyzes OA sources with the multilinear engine (ME-2), providing valuable information for studying the polluted events in Beijing. The manuscript is well written and figures are clearly presented. I recommend it for publication after some minor revisions. Comments: 1. In many sections, the manuscript uses evolution of atmospheric boundary layer (ABL) to explain diurnal and seasonal variations of OA factors. Are ABL measurements available?

[Figure]

It would be helpful if the authors can show ABL results and include a more detailed discussion comparing ABL and OA. 2. Regarding regional transport, the manuscript proposes 200 km as the distance of surrounding regions to control emissions in winter. Have the authors considered influences of different weather systems on regional transport in each season? Also, please provide more details on how '200 km' is determined? Is there any evidence from back trajectory? 3. For 'regional cloud processing' in line 20 on page 14, can the authors elaborate more on this? Also, will cloud convection influence MO-OOA transport? 4. The authors propose explanations for different phenomena, such as 'one explanation is that MO-OOA from a regional scale was circulated from the Bohai sea before arriving at Beijing', as well as some explanations in other paragraphs. Please provide references to justify these explanations. 5. In line 27 on page 15, can the authors explain more about 'the formation of MO-OOA from both aqueous-phase and photochemical processing'? The manuscript also mentions aqueous-phase processes in other paragraphs. It would be helpful if the authors can elaborate more on why MO-OOA and sulfate correlation leads to the conclusion that MO-OOA is formed through aqueous-phase reactions, and also provide some typical aqueous-phase reactions for MO-OOA formation. In addition, can the authors provide some insights on the possible contribution percentages of aqueous-phase processes and photochemical processes. Identifying dominant processes would be very useful for OA studies.

---

## Referee Comment (RC2) · Anonymous Referee #2 · 9 Mar 2018

Review of Sun et al. "Source apportionment of organic aerosol from two-year highly time-resolved measurements by an aerosol chemical speciation monitor in Beijing, China"

The authors present two-year measurements on organic aerosols (OA) in Beijing with an aerosol chemical speciation monitor (ACSM). Source apportionment was performed using multilinear engine (ME-2). Five factors, including fossil fuel organic aerosol (FFOA), biomass burning organic aerosol (BBOA), cooking organic aerosol (COA), less oxidized oxygenated organic aerosol (LO-OOA), and more oxidized oxygenated
organic aerosol (MO-OOA). Based on the source apportionment results, seasonal variations, loading dependence (Section 3.2, see suggestion on change of section title below), RH/T dependence, and potential source regions, were thoroughly discussed in view of emission and formation of OA in Beijing. The dataset is highly valuable in view of its long duration and deep analyses. The analyses are rigorous and the manuscript is generally well written. This work is certainly within the scope of ACP and of interest to readers of ACP. I do, however, have a few points for the authors to address, as below.

Major comments:

1. Mixing of factors. It is understood that statistical models like positive matrix factorization (PMF) suffer from this problem, even with ME-2, and especially with less chemically resolved ACSM. The mixing of hydrocarbon-like organic aerosol (HOA) and coal combustion organic aerosol (CCOA), resulting in FFOA, is a well justified one. But there are still a few other complications. For example, in page 9 line 10 and line 20, the authors admitted that FFOA/COA and BBOA/LO-OOA pairs might not be well resolved. Since the entire paper is based on the quantitative fractionation of OA into those factors, one might be skeptical about how well PMF/ME-2 can separate those distinct primary sources (FFOA/COA) and changing primary/secondary OA (BBOA/LO-OOA). In order not to let readers take those numbers (i.e., mass concentrations and mass fractions of different OA factors) for granted, at lease a caveat has to be mentioned about the potential uncertainties in separating OA factors using ACSM data and PMF analysis.

Better yet, further exploration can provide some quantitative estimates on the uncertainties. Take the FFOA/COA pair for example. FFOA (either as HOA or CCOA) has associated markers (BC and/or Chl) that might be strongly correlated with it (HOA or CCOA) according to previous studies. If using data points from periods with minimal COA contribution (e.g., morning rush hours or after midnight), one can get good correlations and scale factors between FFOA and BC/Chl in non-cooking periods. In the case of FFOA in summer, which is suspiciously mixed with COA, one can use the

scale factors to estimate the "real" FFOA during cooking periods. Although the so-called "real" FFOA is even less rigorously obtained compared to that from PMF/ME-2, one can at least have a quantitative understanding on how much difference can it be between the FFOA obtained from these two methods, serving as an uncertainty for PMF analysis.

2. PSCF analysis. The potential source analysis on primary (assuming unchanged) is reasonable, but might be easily over-interpreted. For instance, the author stated that COA should not be regional, but its "potential source" can extend to very far a distance from the sampling site (to the Bohai Sea in Sp12 and S12 in Figure 8, and distinct hot spots southwest of the city in F11 and Sp13). The same analysis applied to secondary factors might be even more difficult to apprehend. In that type of analysis, does it mean the precursors are from those regions or the oxidation occur in those regions? Again, some precautions should be mentioned to avoid over-interpretation of PSCF results, which only provide a very rough estimate on the coupling of air mass transport and chemical species.

3. Conclusions. With a number of aspects discussed, the manuscript presents a number of conclusions. It is difficult to pin-point what new findings were obtained with this two-year dataset (very valuable, indeed), and how much this work is different from other AMS studies in Beijing, many of which were done by the authors. I strongly suggest the authors to distill the conclusions into one or two major leaps that this work achieves compared to other one-month or even one-year measurements.

4. Some obstacles in smooth reading. There are a number of places that requires careful grammatical check. Below in the minor comments are a few examples I spotted. More thorough checking will help increase the readability of this paper.

Minor comments:

1. Page 2, line 22. "real-time" to "real time".

2. Page 2, line 24. I don't think PCA/PMF/ME-2 can "differentiate" OA factors from sources/processes. They can resolve the OA matrix into different factors, which correspond to different sources/processes.

3. Page 2, line 30-31. Two sentences here that need splitting.

4. Page 3, line 8. "one year" to "one-year".

5. Page 3, line 9. "season variation" to "seasonal variation".

6. Page 3, line 11. "this study". Which study? Sun et al. (2015)? Zhang et al. (2013)? Or Hu et al. (2017)? Are all these three studies lack of seasonal variations of OA?

7. Page 3, line 13. "two years'" to "two-year", and other few places using the same form.

8. Page 4, line 9-11. It is ambiguous here. Should be "...and default relative ionization efficiencies (RIE) were used, except for ammonium whose RIE was determined from measurements with ammonium nitrate."?

9. Page 4, line 18. "mass resolution" to "low mass resolution".

10. Page 5, line 11. "the seasonal ME2-ACSM reports" looks odd. Should be "the seasonally average SOA concentrations are overall 16% higher using PMF/ME-2 analysis with the ACSM dataset compared to those using conventional PMF analysis."?

11. Page 5, line 14. "ME-2 analysis" to "from ME-2 analysis".

12. Page 5, line 15. "during the first eight months' measurements" to during the first eight months".

13. Page 5, line 16. "during the rest of months" to "during the other months"; "with differences less than 3%" to "with differences of less than 3%".

14. Page 6, line 14. "which is much higher" to "which are much higher".

15. Page 7, line 1-2. This can be incorrectly understood as COA is more important

than traffic emissions in all time in Beijing. Suggest to put "in non-heating seasons" to the second half of the sentence.

16. Page 7, line 15. "much differences" to "many differences".

17. Page 8, line 3 – 22. The first and the last sentences of this paragraph looks contradicting. Does MO-OOA have a pronounced seasonal variation or not?

18. Page 9, Section 3.2. Suggest to change the section title to "Loading-dependent OA composition" because it is basically what this section is about.

19. Page 10, line 14. "during lunch and dinner times respectively" to "during lunch and dinner times, respectively".

20. Page 11, line 25. "at ∼50 – 60%" to "to ∼50 – 60%".

21. Page 12, line 17. FFOA emissions or FFOA formation? I believe FFOA is primary (emission).

22. Page 12, line 24. "dependence" to "dependences".

23. Page 14, line 6. "surprising" to "surprisingly".

24. Page 14, line 7. Delete "we".

25. Page 15, line 10. The previous discussion stressed on the constant mass concentrations of COA, not constant mass fractions. And the authors stated previously that COA mass fractions increased during clean periods. Should this be modified to be consistent with the points made in the discussion?

26. Page 15, line 26. "correlated"? The authors used a present tense almost throughout the whole discussion. How come a past tense is used here suddenly? Is it "is correlated"?

27. Page 25, Figure 1-c1. "FOA" to "FFOA".

---

## Author Comment (AC1) · 27 Apr 2018

We are thankful to the two reviewers for their thoughtful comments and suggestions that help improve the manuscript significantly. We have revised the manuscript accordingly. Listed below are our point-by-point responses in blue to each reviewer's comments

**Response to reviewer #1**

Aerosol source apportionment is crucial for employing effective control strategies to improve air quality. This study presents seasonal and diurnal variations of different OA factors based on long-term (2 years) observations, and analyzes OA sources with the multilinear engine (ME-2), providing valuable information for studying the polluted events in Beijing. The manuscript is well written and figures are clearly presented. I recommend it for publication after some minor revisions.

We thank the reviewer's positive comments.

Comments:

1. In many sections, the manuscript uses evolution of atmospheric boundary layer (ABL) to explain diurnal and seasonal variations of OA factors. Are ABL measurements available? It would be helpful if the authors can show ABL results and include a more detailed discussion comparing ABL and OA.

We thank the reviewer's comments. Unfortunately, the measurements of ABL heights were not available in this study. However, the mixing layer heights (MLHs) that were previously retrieved from the ceilometers measurements in Beijing show clear diurnal cycles during all seasons (Tang et al., 2016). As shown in Figure R1, the average MLHs at nighttime was ~400 – 500 m in spring and summer in Beijing, which are ubiquitously lower than those during daytime. Also, MLHs showed clearly daytime increases from ~400 – 500 m in early morning to > 1 km in the late afternoon. Such diurnal variations of MLHs support our conclusions. In the revised manuscript, the reference of Tang et al. (2016) was cited to support our discussions.

[Figure]

Figure R1. Average diurnal cycles of MLHs in spring and summer in Beijing (Tang et al., 2016).

2. Regarding regional transport, the manuscript proposes 200 km as the distance of surrounding regions to control emissions in winter. Have the authors considered influences of different weather systems on regional transport in each season? Also, please provide more details on how '200 km' is determined? Is there any evidence from back trajectory?

Thank the reviewer's comments. We concluded this mainly based on the PSCF analysis during the four seasons. For a reference, the distance between our sampling site (BJ) and Hengshui (HS) city is approximately 250 km. As indicated in Figure 8a, high potential source regions for different OA factors are mostly located in the regions with the distance from Beijing being less than 250 km during all seasons although there are several exceptions (e.g., MO-OOA in Sp13). In the revised manuscript, we corrected 200 km as 250 km.

We agree with the reviewer that the different weather systems can affect the regional transport in each season. For example, the potential source regions in the same season yet in different years, e.g., summer 2011 and 2012, are different. Such differences could be due to the differences in weather systems in the two years. While the analysis of weather systems is beyond the scope of this study, the PSCF in Figure 8 likely suggest two main weather systems from the south/southwest and southeast in different seasons.

3. For 'regional cloud processing' in line 20 on page 14, can the authors elaborate more on this? Also, will cloud convection influence MO-OOA transport?

We thank the reviewer's comments. We originally concluded this because sulfate is dominantly from cloud processing (~80% globally) (Harris et al., 2013) and shows regional characteristics. With the data in this study, it is very difficult to characterize the influence of cloud convection on MO-OOA transport. Still, we reworded this sentence in the revised manuscript as: "with sulfate that is mainly formed over a regional scale (Zhang et al., 2012)".

4. The authors propose explanations for different phenomena, such as 'one explanation is that MO-OOA from a regional scale was circulated from the Bohai sea before arriving at Beijing', as well as some explanations in other paragraphs. Please provide references to justify these explanations.

Such explanations are mainly based on back trajectory analysis. As shown in Figure 8, MO-OOA in summer 2012 shows a high potential source region over Bohai sea. We found that the high concentrations of MO-OOA in summer 2012 mainly occurred during the period of 19 - 21 June when back trajectories show clear transport from Bohai sea to Beijing. This is also consistent with the MODIS image showing severe haze over Bohai sea. Therefore, we conclude that "MO-OOA from a regional scale was circulated from the Bohai sea before arriving at Beijing".

[Figure]

Figure R2. (a) 72 hr back trajectories arriving at Beijing during the period of 19 – 21 June, 2012, (b) the MODIS image on 20 June, 2012 (https://earthdata.nasa.gov/labs/worldview/).

Following the reviewer's suggestions, we added "Indeed, the periods with high concentrations of MO-OOA in summer 2012 (19 – 21 June) show clear transport from Bohai sea to Beijing as indicated by Fig. S9." in the revised manuscript, and also Figure R2 in supplementary.

5. In line 27 on page 15, can the authors explain more about 'the formation of MO-OOA from both aqueous-phase and photochemical processing'? The manuscript also mentions aqueous-phase processes in other paragraphs. It would be helpful if the authors can elaborate more on why MO-OOA and sulfate correlation leads to the conclusion that MO-OOA is formed through aqueous-phase reactions, and also provide some typical aqueous-phase reactions for MO-OOA formation. In addition, can the authors provide some insights on the possible contribution percentages of aqueous-phase processes and photochemical processes. Identifying dominant processes would be very useful for OA studies.

Thank the reviewer's comments. Sulfate is dominantly from aqueous-phase production, mostly cloud processing, therefore, good correlations between sulfate and MO-OOA support their similar processes. Our previous studies also showed that the aqueous-phase related OOA was well correlated with both sulfate and liquid water content (Sun et al., 2016). Compared with sulfate, nitrate particularly in

wintertime, is mostly from photochemical production (Sun et al., 2013). We found that MO-OOA was also correlated with nitrate for most of the time, suggesting the potential contributions from photochemical production. However, the MO-OOA cannot be further separated into two types of OOA representing photochemical and aqueous-phase processing due to the limited sensitivity, and low mass resolution of the Q-ACSM measurements. In fact, we have evaluated the effects of aqueous-phase and photochemical processing on SOA formation and evolution in Beijing using three HR-ToF-AMS datasets (Xu et al., 2017). The results showed that aqueous-phase processing has a dominant impact on the formation of more oxidized SOA (MO−OOA), and the contribution of MO−OOA to OA increases substantially as a function of relative humidity or liquid water content. Therefore, we expect a dominant contribution of MO-OOA from aqueous-phase processing at high RH levels, while more from photochemical processing at low RH levels.

**Response to reviewer #2**

Review of Sun et al. "Source apportionment of organic aerosol from two-year highly time-resolved measurements by an aerosol chemical speciation monitor in Beijing, China" The authors present two-year measurements on organic aerosols (OA) in Beijing with an aerosol chemical speciation monitor (ACSM). Source apportionment was performed using multilinear engine (ME-2). Five factors, including fossil fuel organic aerosol (FFOA), biomass burning organic aerosol (BBOA), cooking organic aerosol (COA), less oxidized oxygenated organic aerosol (LO-OOA), and more oxidized oxygenated organic aerosol (MO-OOA). Based on the source apportionment results, seasonal variations, loading dependence (Section 3.2, see suggestion on change of section title below), RH/T dependence, and potential source regions, were thoroughly discussed in view of emission and formation of OA in Beijing. The dataset is highly valuable in view of its long duration and deep analyses. The analyses are rigorous and the manuscript is generally well written. This work is certainly within the scope of ACP and of interest to readers of ACP. I do, however, have a few points for the authors to address, as below.

We thank the reviewer's positive comments.

Major comments:

1. Mixing of factors. It is understood that statistical models like positive matrix factorization (PMF) suffer from this problem, even with ME-2, and especially with less chemically resolved ACSM. The mixing of hydrocarbon-like organic aerosol (HOA) and coal combustion organic aerosol (CCOA), resulting in FFOA, is a well justified one. But there are still a few other complications. For example, in page 9 line 10 and line 20, the authors admitted that FFOA/COA and BBOA/LO-OOA pairs might not be well resolved. Since the entire paper is based on the quantitative fractionation of OA into those factors, one might be skeptical about how well

PMF/ME-2 can separate those distinct primary sources (FFOA/COA) and changing primary/secondary OA (BBOA/LO-OOA). In order not to let readers take those numbers (i.e., mass concentrations and mass fractions of different OA factors) for granted, at lease a caveat has to be mentioned about the potential uncertainties in separating OA factors using ACSM data and PMF analysis. Better yet, further exploration can provide some quantitative estimates on the uncertainties. Take the FFOA/COA pair for example. FFOA (either as HOA or CCOA) has associated markers (BC and/or Chl) that might be strongly correlated with it (HOA or CCOA) according to previous studies. If using data points from periods with minimal COA contribution (e.g., morning rush hours or after midnight), one can get good correlations and scale factors between FFOA and BC/Chl in non-cooking periods. In the case of FFOA in summer, which is suspiciously mixed with COA, one can use the scale factors to estimate the "real" FFOA during cooking periods. Although the socalled "real" FFOA is even less rigorously obtained compared to that from PMF/ME-2, one can at least have a quantitative understanding on how much difference can it be between the FFOA obtained from these two methods, serving as an uncertainty for PMF analysis.

The reviewer suggested an excellent point for evaluating the uncertainties of ME-2 results. In fact, it is very challenging to separate traffic related HOA from COA using quadrupole AMS or ACSM measurements in summer (Sun et al., 2010;Sun et al., 2012). We then tried to use black carbon (BC) as a tracer to separate HOA from COA assuming that BC is dominantly from traffic emissions while the contribution from cooking is small (Pei et al., 2016). Because BC was not available for the entire study, here we just evaluated the BC-tracer method in August 2012 with two-factor PMF solution, i.e., POA and SOA. We first checked the hourly correlations between BC and POA, and found that these two species were well correlated ($r^2 > 0.75$) between 4:00 – 10:00 when cooking emissions are not significant (Figure R3a). The ratios of POA/BC were also the lowest during this period of time, suggesting the dominant contribution of traffic related HOA to POA.  Figure R3b shows the scatter

plot of POA vs. BC. The average ratio of POA/BC during the period without significant cooking influences is 0.96. With this, we estimated the mass concentrations of HOA and COA as below:

(1) $HOA = (POA/BC)_{nc} \times BC - COA_b$

(2) $COA = POA - HOA$

Where POA is primary organic aerosol from two factor solution of PMF analysis, $(POA/BC)_{nc}$ is the average ratio of POA/BC during the periods without significant influences from cooking, which is 0.96 here, and $COA_b$ is the background concentration of cooking aerosols. According to the diurnal cycles of COA in previous studies in Beijing (Huang et al., 2010;Hu et al., 2016;Hu et al., 2017), we found a background concentration of $1.30 \pm 0.39$ µg m$^{-3}$ even during periods without cooking emissions.

[Figure]

Figure R3. (a) Diurnal variations of POA, BC, correlation coefficients and slopes of POA vs. BC, (b) scatter plot of POA vs. BC, and the red circles are the data between 4:00 – 10:00.

As shown in Figure R4, the estimated time series of COA is relatively well correlated with *m/z* 55, which has a significant contribution from cooking aerosol (Sun et al., 2011), and the diurnal cycle is also consistent with previous observations showing two pronounced peaks during meal times. The estimated HOA and COA on average contributed 12% and 13%, respectively to OA in August 2012. Compared with the results of ME-2 analysis in Table 1, the contribution of COA is the same (13%), while that of HOA (6%) from ME-2 analysis is much lower.

One of the reasons is that the BC-tracer method did not resolve biomass burning OA, which is ubiquitously observed in Beijing during all seasons (Zhang et al., 2008). Still, the sum of HOA and BBOA from ME-2 analysis is 17%, which has a 5% difference compared with that from the BC-tracer method (12%).

[Figure]

Figure R4. Time series and diurnal cycles of estimated COA and HOA using BC as a tracer. The pie chart shows average OA composition for August 2012.

We also compared with previously resolved HOA and COA from PMF analysis of high resolution AMS spectra at another urban site, Peking university, in Beijing (Huang et al., 2010;Hu et al., 2016;Hu et al., 2017). The results showed that HOA and COA on average contributed 11 – 17% and 16 – 21%, respectively to OA, but varied differently among different years. Such results are comparable with those in our study, which are 5 – 22% and 10 – 18% for HOA and COA, respectively considering that the sampling sites are different. These results together suggest that the results from ME-2 analysis are reasonable in terms of relative contributions although the diurnal cycle of HOA in summer is affected by that of COA.

Following the reviewer's suggestions, we expanded the discussions on ME-2 analysis in Section 2.2

We also noticed that the diurnal cycle of FFOA showed similar pronounced mealtime peaks as that of COA in summer although these two factors were forced to separate in ME-2 analysis. To better understand the uncertainties in quantification of FFOA and COA, we estimated COA concentrations using BC as a tracer for the traffic-related FFOA in August 2012. Indeed, POA from the two factor solution of PMF-ACSM was highly correlated ($r^2 > 0.75$) with BC between 4:00 and 10:00 when cooking emissions are small, suggesting the dominant contribution of traffic emissions on BC. The average ratio of POA/BC during this period of time (0.96) was then used to derive the traffic-related FFOA, and COA was estimated as the difference between POA and FFOA. Our results showed that COA estimated using BC-tracer method on average contributed 13% to OA in August which agrees well with that from ME-2 analysis (Table 1), while that of FFOA (12%) was lower than the sum of FFOA and BBOA (17%). These results together suggest that the results of ME-analysis are quite reasonable even in summer when FFOA and COA are difficult to separate.

2. PSCF analysis. The potential source analysis on primary (assuming unchanged) is reasonable, but might be easily over-interpreted. For instance, the author stated that COA should not be regional, but its "potential source" can extend to very far a distance from the sampling site (to the Bohai Sea in Sp12 and S12 in Figure 8, and distinct hot spots southwest of the city in F11 and Sp13). The same analysis applied to secondary factors might be even more difficult to apprehend. In that type of analysis, does it mean the precursors are from those regions or the oxidation occur in those regions? Again, some precautions should be mentioned to avoid over-interpretation of PSCF results, which only provide a very rough estimate on the coupling of air mass transport and chemical species.

We totally agree with the reviewer that PSCF can only provide a very rough estimate of potential source regions. Although most potential source regions of

primary OA, e.g., COA, are reasonable, the hot spots, e.g., Hengshui (HS) and Baoding (BD) regions for COA in Sp12 are more likely from the uncertainties of PSCF analysis, and do not necessarily indicate that they are important sources of COA in Beijing. This is also supported by previous studies showing that COA dropped rapidly outside of major urban areas (Ots et al., 2016). Comparatively, the wide and uninterrupted source regions are often real, e.g., MO-OOA in summer 2012 (S12) showed a high potential source region over Bohai sea. We found that the high concentrations of MO-OOA in S12 mainly occurred during the period of 19 - 21 June when back trajectories show clear transport from Bohai sea to Beijing. This is also consistent with the MODIS image showing severe haze over Bohai sea (Figure R2). Therefore, we conclude that "MO-OOA from a regional scale was circulated from the Bohai sea before arriving at Beijing". Following the reviewer's suggestions, we didn't over interpret the PSCF analysis.

3. Conclusions. With a number of aspects discussed, the manuscript presents a number of conclusions. It is difficult to pin-point what new findings were obtained with this two-year dataset (very valuable, indeed), and how much this work is different from other AMS studies in Beijing, many of which were done by the authors. I strongly suggest the authors to distill the conclusions into one or two major leaps that this work achieves compared to other one-month or even one-year measurements.

We thank the reviewer's comments. In the revised manuscript, we shortened the conclusions substantially.

4. Some obstacles in smooth reading. There are a number of places that requires careful grammatical check. Below in the minor comments are a few examples I spotted. More thorough checking will help increase the readability of this paper.

We thank the reviewer's comments. We went through the revised manuscript, and corrected the grammar mistakes as much as we can.

Minor comments:

1. Page 2, line 22. "real-time" to "real time".

Corrected

2. Page 2, line 24. I don't think PCA/PMF/ME-2 can "differentiate" OA factors from sources/processes. They can resolve the OA matrix into different factors, which correspond to different sources/processes.

Good point. We revised it as "can further resolve various OA factors that correspond to different sources and processes"

3. Page 2, line 30-31. Two sentences here that need splitting.

It was split

4. Page 3, line 8. "one year" to "one-year".

Corrected

5. Page 3, line 9. "season variation" to "seasonal variation".

Corrected

6. Page 3, line 11. "this study". Which study? Sun et al. (2015)? Zhang et al. (2013)? Or Hu et al. (2017)? Are all these three studies lack of seasonal variations of OA?

It refers to Sun et al. (2015). We clarified this in the revised manuscript.

7. Page 3, line 13. "two years"' to "two-year", and other few places using the same form.

Changes as suggested

8. Page 4, line 9-11. It is ambiguous here. Should be ". . .and default relative ionization efficiencies (RIE) were used, except for ammonium whose RIE was determined from measurements with ammonium nitrate."?

Changed as the reviewer suggested. It now reads: "and default relative ionization efficiencies (RIE) were used, except for ammonium whose RIE was determined from measurements of pure ammonium nitrate."

9. Page 4, line 18. "mass resolution" to "low mass resolution".

Corrected

10. Page 5, line 11. "the seasonal ME2-ACSM reports" looks odd. Should be "the seasonally average SOA concentrations are overall 16% higher using PMF/ME-2 analysis with the ACSM dataset compared to those using conventional PMF analysis."?

This sentence was reworded as "the seasonal ME2-ACSM analysis reports an overall 16% higher SOA concentrations than the conventional PMF-ACSM analysis"

11. Page 5, line 14. "ME-2 analysis" to "from ME-2 analysis".

Added

12. Page 5, line 15. "during the first eight months' measurements" to during the first eight months".

Corrected

13. Page 5, line 16. "during the rest of months" to "during the other months"; "with differences less than 3%" to "with differences of less than 3%".

Corrected

14. Page 6, line 14. "which is much higher" to "which are much higher".

Corrected

15. Page 7, line 1-2. This can be incorrectly understood as COA is more important than traffic emissions in all time in Beijing. Suggest to put "in non-heating seasons" to the second half of the sentence.

Added as suggested

16. Page 7, line 15. "much differences" to "many differences".

Corrected

17. Page 8, line 3 – 22. The first and the last sentences of this paragraph looks contradicting. Does MO-OOA have a pronounced seasonal variation or not?

Thank the reviewer for pointing this out. It is not contradict. The first sentence means "seasonal variation of **mass concentration** of MO-OOA" while the last sentence refers to "seasonal variation of **mass fraction** of MO-OOA".

18. Page 9, Section 3.2. Suggest to change the section title to "Loading-dependent OA composition" because it is basically what this section is about.

Good point. Changed as suggested

19. Page 10, line 14. "during lunch and dinner times respectively" to "during lunch and dinner times, respectively".

Changed

20. Page 11, line 25. "at ~50 – 60%" to "to ~50 – 60%".

Corrected

21. Page 12, line 17. FFOA emissions or FFOA formation? I believe FFOA is primary (emission).

This sentence was now revised as "supporting more FFOA emissions, e.g., from coal combustion, at lower $T$ and higher RH levels."

22. Page 12, line 24. "dependence" to "dependences".

Corrected

23. Page 14, line 6. "surprising" to "surprisingly". 24. Page 14, line 7. Delete "we".

Corrected

25. Page 15, line 10. The previous discussion stressed on the constant mass concentrations of COA, not constant mass fractions. And the authors stated previously that COA mass fractions increased during clean periods. Should this be modified to be consistent with the points made in the discussion?

Thanks for the comments. Although the contributions of COA during clean periods are high, they are relatively constant in terms of seasonal averages. Here we refer to "seasonal average contributions of COA are relatively constant". We added "average" in the sentence to clarify this.

26. Page 15, line 26. "correlated"? The authors used a present tense almost throughout the whole discussion. How come a past tense is used here suddenly? Is it "is correlated"?

Right, it is "is also correlated"

27. Page 25, Figure 1-c1. "FOA" to "FFOA".

Corrected

References:

Harris, E., Sinha, B., van Pinxteren, D., Tilgner, A., Fomba, K. W., Schneider, J., Roth, A., Gnauk, T., Fahlbusch, B., Mertes, S., Lee, T., Collett, J., Foley, S., Borrmann, S., Hoppe, P., and Herrmann, H.: Enhanced Role of Transition Metal Ion Catalysis During In-Cloud Oxidation of SO2, Science, 340, 727-730, 10.1126/science.1230911, 2013.

Hu, W., Hu, M., Hu, W., Jimenez, J. L., Yuan, B., Chen, W., Wang, M., Wu, Y., Chen, C., Wang, Z., Peng, J., Zeng, L., and Shao, M.: Chemical composition, sources and aging process of submicron aerosols in Beijing: contrast between summer and winter, J. Geophys. Res., 121, 1955-1977, 10.1002/2015JD024020, 2016.

Hu, W., Hu, M., Hu, W.-W., Zheng, J., Chen, C., Wu, Y., and Guo, S.: Seasonal variations in high time-resolved chemical compositions, sources, and evolution of atmospheric

submicron aerosols in the megacity Beijing, Atmos. Chem. Phys., 17, 9979-10000, 10.5194/acp-17-9979-2017, 2017.

Huang, X. F., He, L. Y., Hu, M., Canagaratna, M. R., Sun, Y., Zhang, Q., Zhu, T., Xue, L., Zeng, L. W., Liu, X. G., Zhang, Y. H., Jayne, J. T., Ng, N. L., and Worsnop, D. R.: Highly time-resolved chemical characterization of atmospheric submicron particles during 2008 Beijing Olympic Games using an Aerodyne High-Resolution Aerosol Mass Spectrometer, Atmos. Chem. Phys., 10, 8933-8945, 10.5194/acp-10-8933-2010, 2010.

Ots, R., Vieno, M., Allan, J. D., Reis, S., Nemitz, E., Young, D. E., Coe, H., Di Marco, C., Detournay, A., Mackenzie, I. A., Green, D. C., and Heal, M. R.: Model simulations of cooking organic aerosol (COA) over the UK using estimates of emissions based on measurements at two sites in London, Atmos. Chem. Phys., 16, 13773-13789, 10.5194/acp-16-13773-2016, 2016.

Pei, B., Cui, H., Liu, H., and Yan, N.: Chemical characteristics of fine particulate matter emitted from commercial cooking, Front. Environ. Sci. Eng., 10, 559-568, 10.1007/s11783-016-0829-y, 2016.

Sun, J., Zhang, Q., Canagaratna, M. R., Zhang, Y., Ng, N. L., Sun, Y., Jayne, J. T., Zhang, X., Zhang, X., and Worsnop, D. R.: Highly time- and size-resolved characterization of submicron aerosol particles in Beijing using an Aerodyne Aerosol Mass Spectrometer, Atmos. Environ., 44, 131-140, 2010.

Sun, Y., Du, W., Fu, P., Wang, Q., Li, J., Ge, X., Zhang, Q., Zhu, C., Ren, L., Xu, W., Zhao, J., Han, T., Worsnop, D., and Wang, Z.: Primary and secondary aerosols in Beijing in winter: sources, variations and processes, Atmos. Chem. Phys., 16, 8309-8329, 10.5194/acp-16-8309-2016, 2016.

Sun, Y. L., Zhang, Q., Schwab, J. J., Demerjian, K. L., Chen, W. N., Bae, M. S., Hung, H. M., Hogrefe, O., Frank, B., Rattigan, O. V., and Lin, Y. C.: Characterization of the sources and processes of organic and inorganic aerosols in New York City with a high-resolution time-of-flight aerosol mass spectrometer, Atmos. Chem. Phys., 11, 1581-1602, 10.5194/acp-11-1581-2011, 2011.

Sun, Y. L., Wang, Z., Dong, H., Yang, T., Li, J., Pan, X., Chen, P., and Jayne, J. T.: Characterization of summer organic and inorganic aerosols in Beijing, China with an Aerosol Chemical Speciation Monitor, Atmos. Environ., 51, 250-259, 10.1016/j.atmosenv.2012.01.013, 2012.

Sun, Y. L., Wang, Z. F., Fu, P. Q., Yang, T., Jiang, Q., Dong, H. B., Li, J., and Jia, J. J.: Aerosol composition, sources and processes during wintertime in Beijing, China, Atmos. Chem. Phys., 13, 4577-4592, 10.5194/acp-13-4577-2013, 2013.

Tang, G., Zhang, J., Zhu, X., Song, T., Münkel, C., Hu, B., Schäfer, K., Liu, Z., Zhang, J., Wang, L., Xin, J., Suppan, P., and Wang, Y.: Mixing layer height and its implications for air pollution over Beijing, China, Atmos. Chem. Phys., 16, 2459-2475, 10.5194/acp-16-2459-2016, 2016.

Xu, W., Han, T., Du, W., Wang, Q., Chen, C., Zhao, J., Zhang, Y., Li, J., Fu, P., Wang, Z., Worsnop, D. R., and Sun, Y.: Effects of Aqueous-phase and Photochemical Processing on Secondary Organic Aerosol Formation and Evolution in Beijing, China, Environ. Sci. Technol., 51, 762–770, 10.1021/acs.est.6b04498, 2017.

Zhang, H., Li, J., Ying, Q., Yu, J. Z., Wu, D., Cheng, Y., He, K., and Jiang, J.: Source apportionment of PM2.5 nitrate and sulfate in China using a source-oriented chemical transport model, Atmos. Environ., 62, 228-242, http://dx.doi.org/10.1016/j.atmosenv.2012.08.014, 2012.

Zhang, T., Claeys, M., Cachier, H., Dong, S., Wang, W., Maenhaut, W., and Liu, X.: Identification and estimation of the biomass burning contribution to Beijing aerosol using levoglucosan as a molecular marker, Atmos. Environ., 42, 7013-7021, DOI: 10.1016/j.atmosenv.2008.04.050, 2008.